# An Updated Focus on Quadruplex Structures as Potential Therapeutic Targets in Cancer

**DOI:** 10.3390/ijms21238900

**Published:** 2020-11-24

**Authors:** Victoria Sanchez-Martin, Carmen Lopez-Pujante, Miguel Soriano-Rodriguez, Jose A. Garcia-Salcedo

**Affiliations:** 1GENYO, Centre for Genomics and Oncological Research, Pfizer/University of Granada/Andalusian Regional Government, PTS Granada, 18016 Granada, Spain; victoria.sanchez@genyo.es (V.S.-M.); carmenlopu@correo.ugr.es (C.L.-P.); 2Microbiology Unit, University Hospital Virgen de las Nieves, Biosanitary Research Institute IBS, Granada, 18014 Granada, Spain; 3Department of Biochemistry, Molecular Biology III and Immunology, University of Granada, 18016 Granada, Spain; 4Centre for Intensive Mediterranean Agrosystems and Agri-food Biotechnology (CIAMBITAL), University of Almeria, 04001 Almeria, Spain

**Keywords:** DNA G-quadruplex, RNA G-quadruplex, i-Motif, cancer

## Abstract

Non-canonical, four-stranded nucleic acids secondary structures are present within regulatory regions in the human genome and transcriptome. To date, these quadruplex structures include both DNA and RNA G-quadruplexes, formed in guanine-rich sequences, and i-Motifs, found in cytosine-rich sequences, as their counterparts. Quadruplexes have been extensively associated with cancer, playing an important role in telomere maintenance and control of genetic expression of several oncogenes and tumor suppressors. Therefore, quadruplex structures are considered attractive molecular targets for cancer therapeutics with novel mechanisms of action. In this review, we provide a general overview about recent research on the implications of quadruplex structures in cancer, firstly gathering together DNA G-quadruplexes, RNA G-quadruplexes as well as DNA i-Motifs.

## 1. Introduction

Besides the Watson–Crick double helix, nucleic acids may adopt alternative secondary structures, such as quadruplex structures (Figure 1). DNA G-rich sequences are able to fold into four-stranded secondary structures known as G-quadruplexes (G4), which arose from the self-association of four guanine bases by Hoogsteen hydrogen bonding within a planar G-tetrad. Self-stacking of two or more G-tetrads generates a G4 structure that is further stabilized by monovalent cations, mainly potassium or sodium, and divalent cations including calcium or magnesium [1,2]. G4s are highly polymorphic intra- or intermolecular structures whose topology can be influenced by variations in strand stoichiometry and polarity, as well as by the nature and length of loops and their location in the sequence [3]. G4 stability is affected by numerous factors, including the number of G-tetrads, loop length, topology and the sequence itself, both within the G4 motif and flanking regions [4]. Direct G4-sequencing in purified single-stranded human DNA identified more than 700,000 G4s [5]. However, in situ mapping by G4 ChIP-seq only detected 1000–10,000 endogenous G4s, which accounted for ~1% of those identified by direct G4-seq, possibly owing to chromatin-associated and other proteins that control the formation of these DNA structures [6]. Endogenous G4s have also been visualized in living cells thanks to G4-specific antibody BG4 and were stabilized by a small-molecule ligand such as pyridostatin [7]. In general, G4s are non-randomly distributed through the genome, being mainly clustered in key regulatory sites such as gene promoters, gene bodies and 5′ untranslated regions (5′UTRs) of highly transcribed genes, particularly those related to cancer, amplification of somatic copy number, ribosomal DNA and telomeres [8]. In these genomic locations, G4s are linked to fundamental biological processes such as transcription, replication, genomic instability and telomeres maintenance [9].

While much attention in the G4 field is focused on DNA, these structures are also present in RNA molecules and are known as RNA G-quadruplexes (RNA G4s). In fact, due to the absence of a complementary strand in RNA molecules, it is widely held that G-rich RNA sequences are more prone to the formation of quadruplex structures. However, the thousands of mammalian RNA regions that can fold into G4s in vitro are overwhelmingly unfolded in cells presumably by robust machinery that globally unfolds RNA G4s [10]. RNA G4s rival their DNA counterparts, often displaying enhanced thermodynamic stabilities [11]. Unlike the highly polymorphic DNA G4s structures that depend on the surrounding conditions, RNA G4s were initially believed to adopt a single conformation. The presence of a 2′-hydroxyl group in the ribose sugar results in additional steric constraints and consequently, the topology of RNA G4s is almost exclusively limited to the parallel conformation [12]. Nevertheless, a recent study has opened avenues to consider that RNA G4s can adopt different conformations apart from the parallel one [13]. Compared to DNA G4s, the detection of cellular RNA G4s has been more challenging. RNA G4s were immunodetected using the same DNA G4 recognizing antibody, BG4. This pioneering study provided substantial evidence for the existence of RNA G4s within the cytoplasm of human cells [14]. In the same study, the selectivity and applicability of carboxypyridostatin, as an example of stabilizing ligand targeting RNA G4s within a cellular context, was corroborated [14]. Furthermore, direct RNA G4-sequencing on poly(A)-enriched RNAs mapped G4s structures in more than 3000 human mRNAs. RNA G4s are mostly found in UTRs, but also in coding sequences. Interestingly, G4s are significantly enriched in micro RNAs (miRNAs) and long noncoding RNAs (lncRNAs), as well as in their target sites, suggesting that G4s may influence the interaction of miRNAs and lncRNAs with their target mRNAs [15]. Moreover, human ribosomal RNA was found to form G4 exposed on the ribosomal surface suggesting potential functions on the recruitment of non-ribosomal proteins and/or polysome assembly [16]. Recently, a novel class of small noncoding RNAs induced by stress and named tiRNAs (tRNA-derived Stress-induced RNAs), which play roles in cancer progression, were reported to form G4 and disturb translational initiation [17]. Overall, RNA G4s emerge as pivotal regulators of pre-mRNA processing, miRNAs maturation, RNA turnover, mRNA targeting and translation [18]. Apart from classical DNA G4s and RNA G4s, a new category of intermolecular G4, named DNA:RNA hybrid G-quadruplex (HQ) has been reported to form when two or three G-tracts are present on the non-template DNA strand downstream the transcription start site (TSS). However, HQs formation has only been validated under in vitro conditions so far [19,20].

C-rich complementary strands always accompany G-rich sequences in genomic DNA. DNA sequences containing C-stretches have been reported to form intercalated, quadruple-helical structures under acidic conditions, being the resulting structure referred to as i-Motif [21]. The tetrameric structure consists of two parallel duplexes combined in an antiparallel manner through the formation of intercalated hemiprotonated C+:C base pairs. These i-Motifs are inserted in different ways to adopt different topologies known as R-, S- and T-forms. Since i-Motifs are formed by hemiprotonated C+:C base pairs, these structures are more stable in a slightly acidic pH and are able to reversibly fold and unfold just by altering the pH. Interestingly, some i-Motif sequences show stable structures even at neutral pH and ambient temperature [22], being favored under conditions of molecular crowding and negative superhelicity [23]. C-rich regions are common within the human genome and its interest has greatly increased in recent years due to its understanding as functional complementary partners to G4s [24,25]. The generation and characterization of an antibody fragment (iMab) that recognized i-Motif structures with high selectivity and affinity has enabled the detection of i-Motifs in the nuclei of human cells [26]. Such structures are formed in key regulatory regions of the human genome, including promoters and telomeric regions, thus implying their involvement in a variety of replication and transcriptional functions [26].

Thanks to the resolution of these non-canonical higher-order structures of nucleic acids and the employment of novel visualization approaches, quadruplex structures have been found to play important roles in several biological events associated with cancer [27]. DNA quadruplexes obstruct the progression of DNA replication forks inducing DNA damage, a hallmark of many tumors. Furthermore, DNA mutations can lead to genomic instability and there is a notable association of quadruplexes with tumoral gene amplification. Lengthening of telomeres is a frequently activated mechanism in cancer and quadruplexes play a key role in telomere biology. Moreover, quadruplexes have been detected in numerous cancer-related genes, interacting with transcription factors or impairing polymerase progression along its template. Thereby, the transcription of different oncogenes and tumor suppressors could be regulated by quadruplex-targeted therapies. In addition, RNA G4s differentially influence translation and mediate recruitment of splicing-associated binding proteins, regulating alternative splicing of numerous important genes in carcinogenesis. Altogether, the cancer-related functions of quadruplexes offer an alternative therapeutic approach in cancer [9,28] (Figure 2).

This review aims to provide an updated report on the literature on quadruplexes regarding cancer therapy. Here we examine the diversity of quadruplexes associated with the six hallmarks of cancer, analyzing their structures and functional implications. Previous reviews have separately covered DNA G4s, RNA G4s and i-Motifs in cancer. Nevertheless, to the best of our knowledge, the present review is the first one gathering together all of these quadruplex structures with relevance in cancer. Since the field of quadruplexes is in continuous change, this review aims to share the current state of knowledge, focusing on the most recent findings in the area.

## 2. Relevant Quadruplex Structures Involved in Cancer

Six vital cellular and microenvironmental processes are considerably de-regulated during oncogenic transformation and malignancy [29]. These distinctive and complementary capabilities include sustaining proliferative signaling, evasion of growth suppressors, resistance to cell death, induction of angiogenesis, and activation of replicative immortality, tissue invasion and metastasis. When each of these hallmarks of cancer is examined, critical genes with a quadruplex structure in the core or proximal promoter are found (Figure 3), and new ones are being continually identified. An overview of quadruplex structures in these cancer-relevant genes is presented in Table 1.

### 2.1. DNA G4s

Multiple G4-containing genes, from oncogenes to tumor suppressors, as well as telomeres and other genomic regions, have been implicated in tumoral processes.

#### 2.1.1. DNA G4s Related to Telomeric Function

Telomeres consist of recurrent TTAGGG-containing sequences playing a crucial role in genomic integrity [88]. These repetitions form an intra/intermolecular antiparallel structure [63]. G4s formation in telomeres is tightly controlled by the Telomere End-Binding Proteins (TEBPs), which resolve telomeric G4s during replication [89]. Telomeric G4s influence the junction of telomeres and the human telomerase reverse transcriptase, hTERT, that contains an RNA template to direct the addition of telomeric DNA and is responsible for regulating telomere lengthening [90]. It has been long assumed that G4s can sequester the 3′ end of the telomere and prevent it from being extended by hTERT [91]. However, hTERT is able to extend parallel, intermolecular G4s conformations in vitro and this ability is conserved among evolutionarily distant species [92]. In fact, recent data reveal that hTERT acts as a parallel G4 resolvase [93]. In addition, hTERT is involved in mitochondrial apoptosis induced by targeted inhibition of *BCL2* [94], regulates the chromatin state and DNA damage response [95] and promotes *c-MYC* and WNT-driven cellular proliferation [96]. Therefore, hTERT is highly expressed in cancer cells and is crucial for limitless replication. The *hTERT* core promoter, approximately from −180 to +1 of its TSS, contains twelve tracts with three or more guanines, which enable the formation of a 68 nucleotide (nt) long G4 and a structure with three stacked parallel units. The formation of this unusual tandem structure disables all three critical binding sites for SP1 transcription factor, thus dramatically downregulating *hTERT* expression and exerting telomere shortening [68].

#### 2.1.2. DNA G4s in Oncogenes

Apart from G4s associated with telomeric function, numerous G4s in oncogenes are involved in a wide variety of processes. *RAS* members (*k-RAS*, *n-RAS*, *h-RAS*) are the most frequently mutated oncogenes and critical drivers of tumorigenesis in pancreatic ductal adenocarcinoma, lung and colorectal cancers [97]. Three distinct types of G4s (near, mid and far) were discovered within the core *k-RAS* promoter. The near 32 nt long *k-RAS* G4 is found −128 nt upstream the TSS and adopts a parallel G4 conformation with a thymidine bulge in one strand and a (1/1/11) looping topology. The mid 52 nt long region from −174 to −226 consists of seven distinct runs of continuous guanines and forms numerous competing isoforms, including a stable three-G-tetrad stacked mixed parallel and antiparallel with loop structures of up to 10 nt. In contrast, the far region from −238 to −273 does not seem to form an inducible and stable structure [71]. G4s presence in *k-RAS* inhibited its transcription, being the mid G4 a stronger repressor of *k-RAS* promoter activity than the near G4 [98]. Recently, a truncated portion of the near G4 was reported to form a parallel head-to-head dimer [99]. With respect to *n-RAS*, there are two G4-motifs near the TSS: one being 28 nt long and positioned in the essential promoter, between −201 and −174, and one being 18 nt long and situated downstream the TSS, between +15 and +32 [39]. With regard to *h-RAS*, it contains two G4-motifs, called *h-RAS1* and *h-RAS2*, which fold into different topologies opening avenues for target selectivity. While *h-RAS1* forms an antiparallel G4, *h-RAS2* forms a parallel structure [39]. These G4s overlap with the binding sites of essential proteins for transcription, disrupting their functioning. The master “undruggable” oncogene *c-MYC,* involved in cell growth, differentiation, proliferation and apoptosis, is overexpressed in a wide variety of human cancers, including most gynecological, breast, small lung carcinomas and colon cancers [100]. Between 85–90% of the transcriptional activation of *c-MYC* is controlled by the nuclear hypersensitivity element III1 (NHE III1), located at position −142 to −115 upstream the P1 promoter [101]. The 27-nt purine-rich strand in the NHE III1 (Pu27) contains around six to ten guanine tracts, adopting an intramolecular parallel-stranded G4 conformation with three G-tetrads and three side loops and acting as a transcriptional repressor element [50,102]. To the same extent, *c-MYB* encodes a transcription factor that plays a critical role in proliferation, differentiation, and survival of hematopoietic progenitor cells. Overexpression of *c-MYB* is involved in the development of some tumors such as human leukemia, colon and breast cancer [103]. Just 17 nt downstream the TSS, there are three copies of a (GGA)_4_ sequence forming an unusual DNA structure that consists of a tetrad/heptad/heptad/tetrad (T:H:H:T) G4 between any two regions of the *c-MYB* GGA repeated sequence. The *c-MYB* G4 region is a critical transcriptional regulatory element and interacts with various nuclear proteins including MAZ, which represses c*-MYB* promoter activity [48]. The oncogene *c-KIT* encodes a tyrosine kinase receptor that stimulates proliferation, differentiation, migration, and survival. Overexpression of *c-KIT* is associated with leukemias, germ cell tumors, different types of mast cell neoplasm gastrointestinal stromal tumors and small cell lung carcinomas [104]. The promoter of *c-KIT* contains two stretches of G-rich tracts, designated as *c-KIT1* and *c-KIT2* [105]. *c-KIT1* is found 87 nt upstream the TSS with a 22-nt sequence consisting of four runs of three guanines, separated by three loop regions (single A residue, a CGCT loop, and an AGGA loop, respectively) forming an all-parallel stranded G4 [106]. On the other hand, *c-KIT2* forms a dimeric G4 which adopts an all-parallel-stranded topology aligned in a 3′ to 5′-end orientation [105]. Both *c-KIT1* and *c-KIT2* flank a 30-nt region, situated 80–101 nt upstream the TSS, which contains a previously unknown all-anti-parallel G4 motif and constitutes a binding site for SP1 transcription factor [45]. Another G4-containing oncogene, *BCL2*, is involved in the regulation of apoptosis and tissue homeostasis and its expression is upregulated in leukemia, ovarian, lung, prostate, breast, colorectal and nasopharyngeal cancer [107]. *BCL2* contains a GC-rich region 1490–1451 nt upstream the P1 promoter, which participates in the regulation of *BCL2* expression since it includes binding sites for SP1, WT1, E2F and NGF transcription factors. The G-rich strand of DNA might adopt any of the three distinct intramolecular structures. The central G4, the most stable one, forms a mixed parallel/antiparallel structure consisting of three G-tetrads connected by loops of one, seven and three nt [108]. A new G4 involving four non successive G-runs was reported to be more stable than the previous central G4, adopting a parallel structure with one 13-nt and two 1-nt chain-reversal loops [34]. Furthermore, early activation of *RET*, encoding a receptor-type tyrosine kinase, is associated with tumorigenesis and thyroid cancer. This gene contains a G-rich strand consisting of five consecutive G-runs, which forms a parallel-type intramolecular G4 in the 3’region of the promoter [56], related to the repression of *RET* transcription [109]. The Hypoxia-Inducible Factor 1-alpha, *HIF1a*, is a master transcriptional regulator that controls oxygen delivery (via angiogenesis) and facilitates metabolic adaptation to hypoxia in cancer cells. *HIF1a* overexpression is related to the progression of brain, breast, cervical, esophageal, oropharyngeal and ovarian tumors [110]. A polypurine-containing tract (−65 to −85) in the promoter of *HIF1a* forms a parallel-unimolecular structure whose mutagenesis diminishes basal *HIF1a* expression [62]. G4s have also been found in CD133, one of the surface markers of cancer stem cells associated with tumorigenicity and metastasis [111]. In particular, *CD133* contains two G-rich sequences within the introns 3 and 7 able to form a parallel and a hybrid G4 respectively, triggering an alternative splicing that dramatically impairs its expression. *CD133* G4s offer new perspectives to target the cancer stem cell subpopulation within the tumoral bulk [43]. Compelling evidence suggests that the Signal Transducer and Activator of Transcription 3 *STAT3* is constitutively activated in many cancers, promoting tumor growth and metastasis [112]. However, depending on the tumoral genetic background, STAT3 might play opposing roles [113]. A G-rich sequence containing four consecutive G-rich tracts is present in the downstream flanking region of the *STAT3* gene, which forms an intramolecular parallel G4 with a downregulation effect [60]. Wilms tumor gene 1 (*WT1*) encodes a zinc-finger transcription factor initially identified as a tumor suppressor gene in Wilms’ tumor, but its overexpression in leukemia has led to consider *WT1* as a potential oncogene. In fact, WT1 increases the expression of *BCL2* and enhances tumor growth and progression to metastatic disease [114,115]. The *WT1* gene promoter contains a G4 with a parallel-intramolecular conformation whose stabilization leads to its transcriptional inhibition [82].

Along with oncogenic transcription factors and transducers, disrupted growth factor signaling contributes to cancer development. Vascular Endothelial Growth Factor (VEGF) comprises a family of endothelial-specific mitogens acting as inductors of angiogenesis and vascular permeability. *VEGF* overexpression, mainly restricted to tumor blood vessels, is responsible for neovascularization in cancer [116,117]. Interestingly, *VEGF* has a promoter region from –85 to –50 upstream the TSS where G4-forming sequences have been identified. The polypurine tract of the *VEGF* promoter consists of five runs of at least three contiguous guanines separated by one or more bases, and displays the characteristic parallel-type signature [118]. The Platelet Derived Growth Factor Subunit A (PDGFa) is a major mitogen for connective tissue cells and other cell types, whose overexpression mediates autocrine stimulation of tumor cells, regulation of interstitial fluid pressure and angiogenesis [119]. A G-rich strand is found in the 5′ region of its promoter (−165 to −139 nt upstream the TSS), which forms two major intramolecular parallel G4s in dynamic equilibrium under physiological conditions [44]. Regarding growth factor receptors, the Vascular Endothelial Growth Factor Receptor 2 (VEGFR2) is functionally relevant in the transduction of pro-angiogenic stimuli incoming from tumor cells [120]. Within the proximal promoter region of *VEGFR2*, there is a G-rich sequence able to form an antiparallel G4, which once efficiently stabilized inhibits *VEGFR2* expression and the angiogenic process [80]. Moreover, the Platelet Derived Growth Factor Receptor Beta (PDGFRb) is essential for cellular growth, proliferation, survival, motility and differentiation and is overexpressed in cancer [121]. The NHE located within the human *PDGFRb* promoter consists of seven G-tracts that can form a mixture of at least four G4s from overlapping sequences. The 5′-mid G4 is the most stable and adopts a primarily parallel intramolecular structure with three 1-nt double-chain-reversal loops and one lateral loop [46]. However, the primary G4 responsible for the repression of *PDGFRb* is located at the 3′-end of the promoter, which has a GGA-containing sequence [47]. Being less stable, the NHE 3′end forms two coexisting intramolecular G4s. One of them has a 3’-non-adjacent flanking guanine inserted into the 3’-external G-tetrad and the other one presents a 5’-non-adjacent flanking guanine inserted into the 5’-external G-tetrad [122]. Finally, the Fibroblast Growth Factor Receptor 2 (FGFR2) controls cellular proliferation, survival and migration acting as a tumorigenic driver [123]. There are three G-rich sequences in the *FGFR2* promoter able to form parallel G4 structures. Strikingly, one of them overlaps the binding site of E2F1, a cis-acting element that regulates *FGFR2* expression [59].

#### 2.1.3. DNA G4s in Tumor Suppressors

There is less evidence showing the presence of G4 structures in the promoter of tumor suppressive genes. This is the case of the retinoblastoma-associated protein RB, a nuclear phosphoprotein that affects the cell cycle and is inactivated in several cancers [124]. An antiparallel intramolecular G4 is present in the G-rich region of the *RB* gene, contributing to double-strand breaks, which could eventually result in destabilization of the *RB* gene [52]. Another example of tumor suppressor is the Poly (ADP-ribose) Polymerase 1 (PARP1) that plays a key role in DNA-damage repair, transcriptional regulation, chromatin remodeling, cell signaling and cell death. Downregulation of PARP1 was found to produce DNA damage and tumorigenesis [125]. A non-canonical G4 with bulges is located at the *PARP1* promoter, 125 nt upstream the TSS. In particular, it forms a (3+1) hybrid, intramolecular, three-layered G4 topology with unique structural features [42].

#### 2.1.4. DNA G4s in Other Genomic Elements

Interestingly, G4s are found in other genomic regions of therapeutic relevance in cancer. DNA deletions in mitochondrial DNA are prevalent in cancer and G-rich sequences near deletion breakpoints form antiparallel inter and intramolecular G4 structures [33]. A direct role regarding G4 disruption in the context of mitochondrial genome replication, transcription and respiratory function has been suggested [126]. Moreover, several putative G4-forming sequences are present in the non-template strand of the ribosomal DNA and some of them folds into parallel G4s, controlling ribosomal RNA synthesis. Since increased ribosomal RNA synthesis is required for tumoral cell to meet energetic demands, G4s in ribosomal DNA are relevant in tumorigenesis [54].

### 2.2. RNA G4s

This section is focused on different G4s implicated in tumorigenesis, including RNA G4s found in telomeric sequences, in UTRs and within splicing sites where G4s control the expression of oncogene and tumor suppressors. Recent discoveries of G4s harbored in miRNAs and lncRNA are also discussed.

#### 2.2.1. RNA G4s Related to Telomeric Function

Similar to DNA G4s, G4s are also found in telomeric RNA. The ~100- to 9000-nt telomeric repeat-containing RNA, *TERRA*, involved in cellular regulatory functions and chromatin remodeling, contains G4 motifs. Originally, *TERRA* was proposed to adopt an stable and all-parallel conformation with a “beads-on-a-string” like arrangement, whereby each bead was made up of either four or eight UUAGGG repeats in a one- or two-block G4 arrangement, respectively [127]. In contrast, a newly described antiparallel topology in *TERRA* has shown that G4s could also be polymorphic and adopt different structures apart from the canonical parallel configuration [13]. G4s are also found in the transcript of the Telomerase Reverse Transcriptase gene (*hTERT)*, both at splicing sites and at the 5′-terminal region. In particular, spliced intron 6 forming the beta *hTERT* transcript contains several tracks of G-rich sequences able to form G4s. The consequent alteration of the *hTERT* splicing pattern triggers a downregulation of *hTERT* activity [128]. The 18 nt long 5′-terminal region of the *hTERT* RNA also forms a G4 with two subunits, each being a three-layered parallel-stranded G4 with a cytosine bulge. The formation of this stacked dimeric G4 is biologically relevant for its dimerization and other interactions of the active *hTERT* [69]. Moreover, within the Shelterin complex, the Telomere Repeat–binding Factor 2, TRF2, is also implicated in telomere maintenance and overexpressed along tumorigenesis. *TRF2* contains a G-rich RNA sequence located at the 5′UTR, which adopts a stable intramolecular G4 and controls *TRF2* expression by translational repression [74].

#### 2.2.2. RNA G4s in UTRs

Structures of G4s are present in the UTR regions of some of the aforementioned cancer-relevant genes. Most of G4s are found in the 5′UTR region, immediately upstream the initiation codon, influencing post-transcriptional regulation of gene expression. Such is the case of the *RAS* family members (including *h-RAS*, *k-RAS* and *n-RAS*), whose 5′UTR contains several putative G4 sequences according to bioinformatic analysis [39]. The first description of translational repression exerted by a G4 was the highly conserved and thermodynamically stable G4 found in the 5′UTR region of *n-RAS* [129]. Interestingly, when using mRNA reporter constructs that contained the wild-type *n-RAS* G4 located at different positions within the *n-RAS* 5′UTR, the G4 inhibitory effect was found to be dependent on its location: when it was situated within the first 50 nt from the 5′end, this G4 repressed translation, while when it was located at longer distance it showed no effect on translation [40]. The *k-RAS* transcript, characterized by a 192 nt 5′UTR, contains repetitive runs of two guanines which folds in several RNA G4s that repress its translation [72]. Another example is a highly conserved 25-nt G-rich sequence in the 5′UTR of *BCL2* transcript, 42 nt upstream the TSS, with the ability to fold into a G4 and negatively regulates gene translation in vitro [35]. Additionally, the 5′UTR region of the *VEGF* mRNA is an unusually long (1038 nt) GC rich sequence, which harbors two separate internal ribosomal entry sites (IRES), being capable of independently initiating translation with no need of the 5′cap. Interestingly, in *VEGF* IRES, a “switchable” 17-nt sequence containing more than four G-stretches may adopt multiple G4 structures critical for the initiation of *VEGF* cap-independent translation [77,130].

Along with genes harboring both DNA G4 and RNA G4s, other genes of utmost importance in cancer exclusively contain RNA G4s. High expression levels of Cyclin D3 CCND3, observed in several types of cancer, promote the G1/S phase transition in the cell cycle. A G4 folded in an extremely stable, intramolecular, parallel structure is found in the 5′UTR region of the *CCND3* mRNA, and it has been shown to inhibit translation [41]. Additionally, the Transforming Growth Factor-Beta 2 (TGFb2) is a versatile cytokine with a prominent role in cell migration, invasion, cellular development and immunomodulation and it promotes tumor malignancy. In the *TGFb2* 5′UTR region, a 23-nt G-stretch folds into a highly stable intramolecular parallel G4. Intriguingly, unlike the vast majority of G4s located at 5′UTR that functions as translational repressors, the G4 in *TGFb2* displays an activating role in modulating gene expression [67]. Many cancers are estrogen-sensitive with neoplastic growth stimulated through the estrogen receptor, ESR1, a transcription factor that regulates developmental genes. Potential G4-forming sequences are abundant in the human *ESR1* gene, especially within exonic regions where three out of a total of twenty have been identified. In particular, a parallel G4 is present at the 5′UTR exon C- derived mRNA, acting as a modulator of genetic translation [55]. Additionally, BAG Cochaperone 1 (*BAG1*) encodes an anti-apoptotic protein whose overexpression suppresses the activation of apoptotic caspases. The 5’UTR region of *BAG1* mRNA contains a G4 which exerts a repressive effect on its cap-dependent translation and, conversely, a favorable effect on its cap-independent translation, making this structure a potential target to modulate tumoral phenotypes [32]. Another gene with a G-rich 5′UTR is Ying Yang 1 (*YY1*), which encodes a multifunctional transcription factor belonging to the GLI-Kruppel class of zinc finger proteins. YY1 plays an oncogenic and proliferative role owing to its involvement in the expression of numerous genes mostly involved in cancer [131]. The presence of a mixture of parallel and antiparallel G4 strands has been demonstrated in the promoter and in the 5′-UTR of *YY1,* therefore controlling tumor invasion. Such G4 stacks are complementary to C-rich sequences capable of forming i-Motifs, being both modulators of *YY1* gene expression [84]. Matrix metalloproteinase MT3-MMP, involved in the regulation of cancer cell invasion and metastasis, is overexpressed in the most aggressive nodular-type tumors. The *MT3-MMP* mRNA contains a 20-nt G-rich region upstream the initiation codon, forming an extremely stable intramolecular G4 which has an inhibitory role on translation [31]. Moreover, another metalloproteinase, ADAM10, is implicated in inflammation and cancer invasion [132]. A G4 motif within the 5′UTR mRNA is detected with an inhibitory effect on *ADAM10* translation [30]. Furthermore, the chemokine C-X-C motif ligand 14 (CXCL14) exerts paradoxical roles on tumorigenesis, since it suppresses the in vivo growth of lung and head-and-neck carcinoma cells while promoting invasion of breast and prostate cancer cells [133]. Five G-runs adopting a parallel G4 structure have been found at its 5′UTR, being implicated in the regulation of *CXCL14* translation [53]. The transcription factor ZIC1, a member of the Zinc finger of the cerebellum (ZIC) protein family, is involved in the inhibition of cell growth and alters the expression of potential target genes in carcinogenesis. Thus, ZIC1 downregulation participates in the progression of human cancer [134]. A 73 nt fragment in the *ZIC1* 5′UTR mRNA folds into a parallel intramolecular G4 and inhibits its translation in vivo [86]. The 5′UTR region of *TAOK2*, which encodes a protein kinase participating in cell signaling and apoptosis induction [135], folds into a parallel G4 in vitro and decrease the efficacy of *TAOK2* translation. Intriguingly, an adenine to guanine mutation observed in some cancer patients adds an extra G-run to the existing four G-runs. Such mutation further stabilizes the G4 structure changing the arrangement of the G-runs and shortening one of the loops [53]. Another tumor suppressor gene, the Hepatocyte Nuclear Factor 4 alpha *HNF4a*, plays a key role in the repression of promitogenic genes, the crosstalk with other cell cycle regulators and the regulation of miRNAs. Inhibition or loss of *HNF4a* promotes tumorigenesis and its re-expression results in decreased cancer growth [136]. A classic parallel G4 motif at the *HNF4a* 5′UTR mRNA is required and sufficient to mediate a strong *HNF4a* translational repression [65].

Finally, RNA G4 structures are also present in 3′UTR regions where they can interfere with miRNA binding [137]. In this regard, only the protein kinase PIM-1 has emerged as a relevant example in cancer. *PIM1* controls cell survival, proliferation, differentiation and apoptosis and its overexpression promotes tumorigenesis [138]. A G-rich sequence 277-nt downstream the stop codon at the 3’UTR of human *PIM1* mRNA forms a stable, parallel and intramolecular G4 structure able to inhibit its translation. This is the first evidence of a G4 located in the 3’UTR region acting as a post-transcriptional regulator [49]. Recently, a G4 structure has been discovered in the 3′UTR region of Long Interspersed Element class 1 (*LINE-1* or *L1*) retrotranposon, stimulating retrotransposition [75]. Such transposition and recombination events contribute to genomic instability and participate in cancer development [139].

#### 2.2.3. RNA G4s in Splicing Sites

The *TP53* gene encodes the tumor protein 53 (or p53), the well-known guardian of the genome, which is frequently mutated in human cancers [140]. Several G4s located in *TP53* intron 3 affect the intron 2 splicing, leading to differential expression of transcripts encoding distinct TP53 isoforms [70]. Another G4 located in the vicinity of a polyadenylation site in *TP53* allows the transcript to be properly processed [141,142]. Moreover, BCLX, a member of the BCL2 family, is a mitochondrial transmembrane protein that regulates the intrinsic apoptotic pathway. An alternative splicing event from two possible 5′ splicing sites in exon 2 of *BCLX* results in two isoforms with antagonistic effects on cell survival. The long isoform, BCLX-L, is the most abundant and inhibits apoptosis. In contrast, the short isoform BCLX-S directly binds to and inhibits the anti-apoptotic BCLX-L and BCL2 proteins by forming heterodimers. Multiple splicing factors influence the *BCLX* splicing ratio, which is involved in esophageal cancer, dysplasias and cell carcinomas [143]. Interestingly, G-rich sequences have been found near *BCLX* alternative splicing sites, folding into parallel G4s. Particularly, one of them is located upstream the *BCLX-S* splicing site, four between the *BCLX-S* and the *BCLX-L* splicing sites, and one downstream the 3′ splicing site [38,144]. The last G4 structure included in this group is found in the EWS RNA binding protein 1, *EWSR1*, one of the genes most commonly involved in sarcoma translocations, rendering novel transcription factors with tumoral effects [145]. *EWSR1* mRNA contains a parallel-tetramolecular G4 structure within exon 8, which enables the recruitment of HNRNPH1, a component of the heterogeneous nuclear ribonucleoprotein (hnRNP) complex required for the processing of distinct *EWSR1* transcript variants [57].

#### 2.2.4. RNA G4s in Non-Coding RNAs

RNA G4 structures have been found in tumorigenesis-involved non-coding RNAs, including lncRNAs and miRNAs [146,147]. The recently described lncRNA, *GSEC*, harbors a G4 in its sequence from nucleotides 11 to 26, which allows *GSEC* binding to RNA helicase DHX36 and its consequent inhibition. *GSEC* is relevant in cancer since it is upregulated in tumoral cells and is required for their migration [61]. Moreover, different G-rich regions have been registered within miRNAs. As a rule, the double-stranded RNA endoribonuclease Dicer recognizes canonical stem–loop structures in pre-miRNA to produce mature miRNAs, but G4s found in these regions mediate deviation to deregulated levels of mature miRNA [148]. In particular, the oncogenic pre-miRNA-92b, implicated in the control of cellular growth by directly targeting the tumor suppressor *PTEN* [149], contains six G-stretches with three guanines located 2–28 nt from the 5′ end. This sequence forms a very stable G4 covering about half the length of pre-miRNA-92b stem-loop. Consequently, the G4 potentially destabilizes the stem-loop structure, impairing Dicer-mediated maturation both in vitro and in vivo [87]. Furthermore, miRNA-1587, a miRNA secreted by exosomes from glioma stem cells, targets and induces downregulation of the tumor suppressor *NCOR*, promoting tumorigenesis [150]. Under a high concentration of ammonium or a molecular crowding environment, a stable parallel G4 in miRNA-1587 folds into a dimeric form through 3′-to−3′ stacking of two monomeric G4 subunits. This structure is implicated in the regulation of miRNA-1587 expression [81]. Another miRNA, miRNA-3620-5p, shares the same function as miRNA-1587, increasing the proliferation of glioma stem cells [150]. Moreover, miRNA-3620-5p is a master regulator in the TP53-mediated network [151]. It displays a stable pattern of a three-layered parallel G4 with three loops and two flanking bases at each end [85]. Among the different functions of G4s found in tumor suppressive miRNAs, the low expression of miRNA-26a in human cancers [152] derives from the effect of a G4 located 23–48 nt from the 5′end. This G4 impairs the pre-miRNA-26a maturation and decreases its expression. The structure is formed by four G-stretches, two guanines in the first three G-tracts and three guanines in the last G-tract, in line with a non-canonical G4 with two G-tetrads [83]. An onco-suppressive role was also attributed to miRNA-149 controlling cell migration and apoptosis with a low expression in cancer cells [153,154]. This miRNA contains a very stable parallel G4 with a total loop length of four bases (1-1-2), which impairs its maturation in cancer cells [79].

### 2.3. i-Motifs

The formation of a G4 in the genomic DNA leaves the complementary C-rich sequence single-stranded, which might therefore fold into an i-Motif. Consequently, the presence of G4s or their residues enhances i-Motifs stability [155]. However, the formation of G4s and i-Motifs in the two complementary strands is mutually exclusive in a variety of DNA templates [156]. How cells coordinate the formation of these two quadruplex structures with sometimes conflicting biological activities still remains unclear. Despite the fact that quadruplex i-Motifs have been found both in telomeric and extra-telomeric regions with a notorious involvement in cancer, i-Motifs have been far less studied than G4s.

#### 2.3.1. Telomeric i-Motifs

Under near-physiological conditions of pH, temperature and salt concentration, telomeric DNA is predominantly in a double-helix form. However, at lower pH values or higher temperatures, G4s and/or i-Motifs efficiently compete with the duplex [64,157]. Much work has been done to understand the behavior of telomeric i-Motifs in different environments. C-rich telomeric sequences at the 3’end are able to fold into an i-Motif through intercalated C:C+ base pairs in a variety of experimental conditions [158]. Human telomeric i-Motifs persist folded even at ambient temperature and neutral pH [22]. However, natural base lesions marginally disturb the formation of these i-Motifs at telomeric sites [159]. In addition, cytosines in a C-rich DNA sequence act as major epigenetic targets for methylation. DNA i-Motifs are stabilized when modified with one or two 5-methylcytosines, but hypermethylation with 5-methylcytosines and single modification with 5-hydroxymethylcytosine cause destabilization of the structure [160].

#### 2.3.2. Extra-Telomeric i-Motifs

C-tracts can be found in the promoter of several oncogenes; such is the case of three tandem i-Motif-forming sequences in *k-RAS* promoter, complementary to the three previously described G4 regions. In particular, the i-Motif formed in the C-rich mid region is the most stable and exists in a dynamic equilibrium with hairpin species. The transcription factor hnRNPK selectively binds to these i-Motifs and positively modulates *k-RAS* transcription [73]. Another *RAS-*member, *h-RAS*, is also regulated by a G4/i-Motif switch, which interacts with proteins that recognize non-B DNA conformations. In this regard, the two C-rich regions of *h-RAS* promoter fold into two i-Motifs, which are tightly bound by the heterogeneous nuclear ribonucleoprotein A1 (hnRNPA1). This ribonucleoprotein unfolds the i-Motif structures upon binding, which leads to the activation of *h-RAS* expression [66]. In addition, C-rich sequences are found in the promoter of *c-MYC,* cooperatively forming an intramolecular fold-back i-Motif [51]. Nevertheless, the folding process is not a simple two-state transition. Instead, it involves a partially folded conformation as an intermediate state in which the bases are not as efficiently stacked as in the completely folded i-Motif form [161]. The first intron of *n-MYC* contains a 12-nt C-rich tract which forms two i-Motifs, differing in the protonation of the bases located at the loops. A stable Watson–Crick hairpin is formed by the bases in the first loop, stabilizing the i-Motif that coexists with G4s in a broad range of pH [37]. The antiapoptotic protein BCL2 contains an i-Motif in its gene. A C-rich sequence is found directly upstream the *BCL2* promoter, being capable of forming one major intramolecular i-Motif with a predominant 8:5:7 loop conformation. This region was shown to be essential to modulate *BCL2* promoter activity, as deletion or mutation of this region significantly increased transcription [36]. The protooncogen *RET*, in addition to the previously mentioned polypurine sequences in the 3’ region of its promoter, also contains polypyrimidine-rich sequences able to form an i-Motif which modulates *RET* basal transcription. In a pH-dependent manner, 17 nt in the C-rich strand are predicted to form five C:C+ base pairs and three loops (2:3:2 loop sizes) in the i-Motif structure [56]. Moreover, *VEGF* promoter is able to form an intramolecular antiparallel i-Motif structure that involves six C:C+ base pairs and a 2:3:2 loop configuration, playing a key role in *VEGF* transcriptional activation [78]. Another growth factor, *PDGFRb*, shows at least two different i-Motifs formed through the differential use of C-runs in the NHE. The formation of these i-Motifs and complementary G4s in *PDGFRb* was found to be sensitive to point mutations, which shifted the dynamic equilibrium to favor different transcriptional status [47].

Along with i-Motifs in oncogenes, C-rich regions fold into i-Motif structures within the promoter of tumor suppressor genes such as *RB* and *SMARCA4*. In *RB*, an 18-nt region folds into an intramolecular i-Motif structure, competing with the double-helix at low pH values [52]. Additionally, the ATP-dependent chromatin remodeler *SMARCA4*, frequently mutated and downexpressed in ovarian and lung cancer [162], contains a C-rich region from −71 to −28 nt upstream of its promoter. Structurally, this region contains one tract with four cytosines, three tracts with three, and two tracts with two, forming a relatively stable and homogeneous intramolecular i-Motif in terms of both pH and temperature [58]. Future studies might reveal if i-Motifs in *RB* and *SMARCA4* could provide new mechanisms for the modulation of gene expression.

## 3. Therapeutic Relevance of Quadruplex Structures in Cancer

Quadruplex structures offer a new modality for targeting DNA and RNA and new insights are put into quadruplexes for the treatment of cancer [27]. The distinct molecular features of the distinct quadruplexes enable structure-selective recognition by small molecules [163]. However, whereas early studies focused on modulating individual cancer genes, the prevalence of quadruplexes in many cancer-promoting genes suggests that collectively targeting multiple quadruplexes (thus altering the expression of many such genes) would be a feasible strategy [164].

To date, thousands of small-molecules binding to quadruplex structures have been reported in the G-Quadruplex Ligands Database [165]. G4 binders generally have an aromatic surface for π-π stacking with G-tetrads, a positive charge or basic groups to bind to loops or grooves of the G4, and steric bulk to prevent intercalation with double-stranded DNA [166]. Several quadruplex ligands have been evaluated for their therapeutic potential as a novel anti-cancer strategy and have shown antitumoral activity in vitro and through xenograft models [163]. For instance, the well-known cationic porphyrin TMPyP4 acts as a general stabilizer of quadruplexes including DNA [167] and RNA G4s [168], but also i-Motifs [169]. The same occurs with the pan-quadruplex binding molecule, Pyridostatin, which binds to all quadruplex structures [57,170,171]. In the case of BRACO-19, it shows binding affinity for both DNA G4s [172] and i-Motifs [170]. However, selectivity is gaining momentum in the last years and has resulted to be outstanding for two peptidomimetic ligands (PBP1 and PBP2). Whereas the para-isomer PBP1 exhibits high selectivity for *BCL2* i-Motif, the meta-isomer PBP2 selectively binds to *BCL2* DNA G4. Consequently, PBP1 upregulates *BCL2* gene expression and PBP2 downregulates it in cancer cells [173]. Some quadruplex ligands are even in clinical trials. Of particular note is CX-5461, which has recently entered clinical trials for patients with BRCA-deficient tumors [174]. CX-3543, also named quarfloxin, passed Phase II trials as a candidate therapeutic agent against several tumors, but Phase III trials were not completed due to its high binding to albumin [175]. The chemical structures of these quadruplex ligands and their associated antitumoral activities are included in Table 2.

## 4. Future Perspectives and Conclusions

Recent advances in quadruplexes detection have provided a substantial body of new data that supports the existence of these structures in the genome of human cells. Although there is much more to be understood about the underlying mechanisms of quadruplex functionality, advances over the last years have corroborated that imbalance in quadruplex dynamics contribute to cancer development. In this regard, quadruplexes have emerged as promising drug targets in antitumor drug discovery. In fact, a series of well-known quadruplex stabilizers have already been demonstrated to possess antitumoral activities in vitro and *in vivo*. Some of them are even undergoing clinical trials. Such is the therapeutic potential of quadruplex targeting that we daringly anticipate that the future treatment of cancer with quadruplex ligands could not be too far away. The main advantage derived from quadruplex-targeted therapies is that some important cancer-relevant genes could be targeted regardless of the druggability of the gene product. Moreover, there is little likelihood of point mutations and emerging resistances. Furthermore, the potential of unique sequence and structures for a given quadruplex would allow drug selectivity by rational design. However, three-dimensional structures for some quadruplexes remain to be determined and new quadruplex targets are continuously being identified. The major limitation impeding the clinical application of quadruplex targeting rests with the selectivity of existing quadruplex ligands. Therefore, more efforts should be devoted to characterize the structural differences between the different quadruplex targets in an attempt to improve the selectivity of ligands. Nonetheless, we are definitely aware that we are just in the beginning of a new “quadruplex era” for cancer treatment.

In this review we have highlighted and discussed biological implications of cancer-relevant quadruplex structures, including DNA G4s, RNA G4s and i-Motifs. The reported quadruplex-mediated antitumoral effects may pave the way for cutting-edge therapeutic approaches in the future treatment of human cancer.

## Figures and Tables

**Figure 1 ijms-21-08900-f001:**
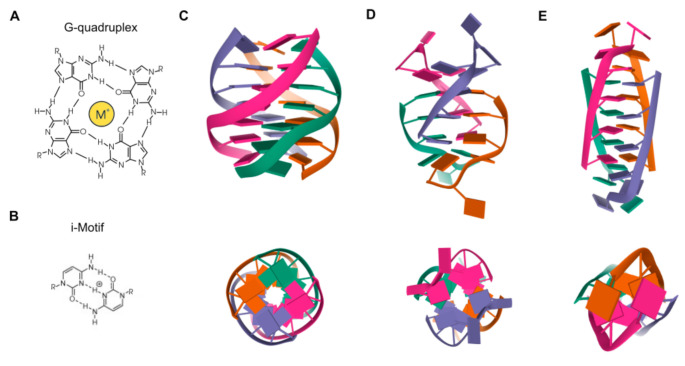
Quadruplex structures. (**A**) Chemical structure of a G4. (**B**) Chemical structure of an i-Motif. (**C**) NMR solution structure of a tetrameric parallel G4 (Protein Data Bank: 139D) from different rotation angles. (**D**) NMR solution structure of an alternating antiparallel tetrameric G4 (Protein Data Bank: 6IMS) from different rotation angles. (**E**) NMR solution structure of a tetrameric i-Motif (Protein Data Bank: 1YBL) from different rotation angles.

**Figure 2 ijms-21-08900-f002:**
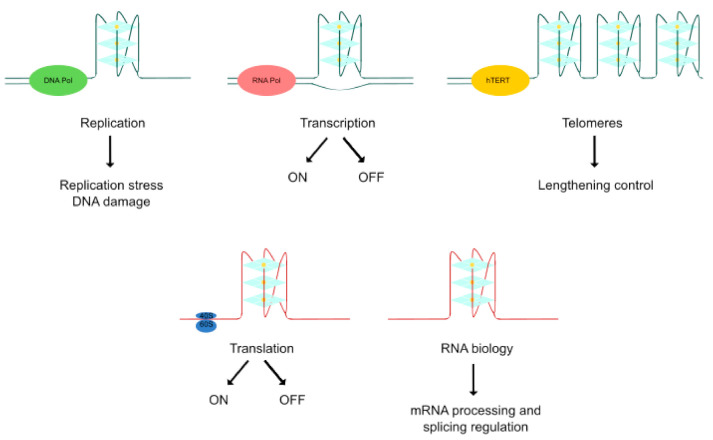
Biological effects of quadruplex structures. The cancer-related functions of quadruplexes derive from their key roles in replication, transcription, lengthening control of telomeres, translation and RNA biology control.

**Figure 3 ijms-21-08900-f003:**
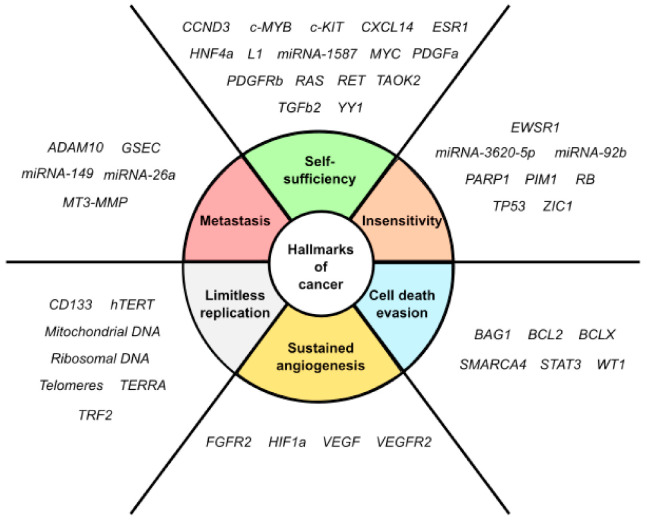
Genes containing quadruplex structures are associated with cancer hallmarks. As described in the text, DNA G4s, RNA G4s and i-Motifs are present in multiple cancer-relevant genes playing key regulatory functions. To facilitate visualization, each gene is only associated with one feature, even though they might be involved in several cancer hallmarks.

**Table 1 ijms-21-08900-t001:** Quadruplex structures in cancer-relevant genes. Schematic representation of all quadruplex structures in cancer-relevant genes with key roles in tumorigenesis, and their topologies. “Not specified” indicates that a quadruplex structure exists but its topology is undetermined. “Mixed” indicates that both parallel and antiparallel topologies are displayed.

Gene	DNA G4	RNA G4	i-Motif	Gene	DNA G4	RNA G4	i-Motif
ADAM10	No	Not specified [30]	No	MT3-MMP	No	Not specified [31]	No
BAG1	No	Not specified [32]	No	Mitochondrial	Antiparallel [33]	No	No
BCL2	Mixed [34]	Not specified [35]	Not specified [36]	n-MYC	No	No	Not specified [37]
BCLX	No	Parallel [38]	No	n-RAS	Not specified [39]	Not specified [40]	No
CCND3	No	Parallel [41]	No	PARP1	3+1 Hybrid [42]	No	No
CD133	Mixed [43]	No	No	PDGFa	Parallel [44]	No	No
c-KIT	Mixed [45]	No	No	PDGFRb	Parallel [46]	No	Not specified [47]
c-MYB	Tetrad:heptad [48]	No	No	PIM1	No	Parallel [49]	No
c-MYC	Parallel [50]	No	Fold-back [51]	RB	Antiparallel [52]	No	Not specified [52]
CXCL14	No	Parallel [53]	No	Ribosomal	Parallel [54]	No	No
ESR1	No	Parallel [55]	No	RET	Parallel [56]	No	Not specified [56]
EWSR1	No	Parallel [57]	No	SMARCA4	No	No	Not specified [58]
FGFR2	Parallel [59]	No	No	STAT3	Parallel [60]	No	No
GSEC	No	Not specified [61]	No	TAOK2	No	Parallel [53]	No
HIF1a	Parallel [62]	No	No	Telomeres	Antiparallel [63]	No	Not specified [64]
HNF4a	No	Parallel [65]	No	TERRA	No	Mixed [13]	No
h-RAS	Mixed [39]	Not specified [39]	Not specified [66]	TGFb2	No	Parallel [67]	No
hTERT	Parallel [68]	Parallel [69]	No	TP53	No	Not specified [70]	No
k-RAS	Mixed [71]	Not specified [72]	Not specified [73]	TRF2	No	Not specified [74]	No
L1	No	Not specified [75]	No	VEGF	Parallel [76]	Not specified [77]	Antiparallel [78]
miRNA-149	No	Parallel [79]	No	VEGFR2	Antiparallel [80]	No	No
miRNA-1587	No	Parallel [81]	No	WT1	Parallel [82]	No	No
miRNA-26a	No	Not specified [83]	No	YY1	No	Mixed [84]	Not specified [84]
miRNA-3620-5p	No	Parallel [85]	No	ZIC1	No	Parallel [86]	No
miRNA-92b	No	Not specified [87]	No				

**Table 2 ijms-21-08900-t002:** Quadruplex ligands. Chemical structure of quadruplex ligands with their targets and antitumoral effects.

Compound	Quadruplex Target	Antitumoral Effect
BRACO-19 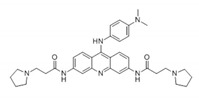	Telomeric DNA G4s	Interference with hTERT [172]
	Telomeric i-Motifs	Interference with hTERT [170]
CX-3543 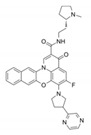	Ribosomal DNA G4s	Inhibition of RNA Polymerase I [175]
CX-5461 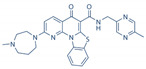	Multiple DNA G4s	Synthetic lethality in BRCA deficient tumors [174]
PBP1, PBP2 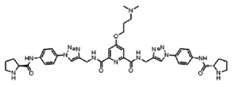	*BCL2* DNA G4 (PBP2)	Downregulate *BCL2* expression [173]
	*BCL2* i-Motif (PBP1)	Upregulate *BCL2* expression [173]
Pyridostatin 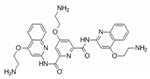	Multiple DNA G4s	Growth arrest by inducing DNA damage [171]
	RNA G4	Regulates alternative splicing of *EWSR1* [57]
	Telomeric i-Motif	Interference with hTERT [170]
TMPyP4 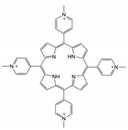	Telomeric DNA G4	Interference with hTERT [167]
	Telomeric RNA G4	Interference with hTERT [168]
	Telomeric i-Motif	Interference with hTERT [169]

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
