# Peer review of "An Updated Focus on Quadruplex Structures as Potential Therapeutic Targets in Cancer"

_ijms, 2020, doi:10.3390/ijms21238900_

Round 1

Reviewer 1 Report

The manuscript “An updated focus on quadruplex structures as potential therapeutic targets in cancer” by Victoria Sanchez-Martin et al. provide a summary of the state of the art in the quadruplex field to the development of new anticancer therapeutics. The authors start to introduce the importance of quadruplex structures in cancer-associated events emphasizing the DNA/RNA G4 structures and i-motifs.

Overall, I support the publication, however, I have some comments/suggestions that I believe would improve this manuscript.

- The review covers the field of quadruplex structures as therapeutic targets in cancer, however, there are many existing reviews in the literature of quadruplex structures, of high quality. The new publication needs to demonstrate that it provides new insights into this field. Some sections already revised by other authors are exhaustively discussed in the manuscript, such as DNA G4s in oncogenes.

-The authors summarize quadruplex structures involved in cancer, but the impact of the review could be increased if authors interpreted/explore more the cancer-related function of quadruplex structures. The review quality also could be increase with the addition of some representative images to illustrate the biological functions of quadruplexes related structures as well as the introduction of high-resolution quadruplex structures available in PDB (kRAS, for example).

- Considering that authors refer to quadruplex structures as potential therapeutic targets, would be interesting a new section to emphasize the clinical importance and applicability of these structures. This section could also provide some information about new quadruplexes ligands (with a table and structure) and its function in the development of cancer.  

- The conclusion section is rather short. This section should provide new future perspectives in this field.

Minor:

The acronym of i-motif (iM) along the text it is not correct according literature. Please consider writing in your long form.

Line 33: G4 are also stabilized by divalent cations. Insert it.

Line 45: are “gene promoters” and not just promoters.

Line 69: insert the correct definition of tiRNAs (tRNA-derived Stress-induced RNAs)

Line 73: switch “to” for “or”

Line 78: i-motifs forms in specific pH values conditions. Acidic conditions are not the only requirement to form these non-canonical structures. Additionally, i-motif could form in neutral pH (See doi: 10.3389/fchem.2020.00040)

Line 87: insert the name of the antibody fragment that recognizes i-motif and respective reference.

Figure 1: please consider some suggestions:

  • Consider placing only a tetrad representation to both DNA and RNA since it is the same in both. RNA only differ of DNA, as is reviewed, in 2’-hydroxyl group of pentose.
  • Increase color contrast
  • Rotate the image because loops cause some confusion specially in hybrid topology

Line 215, please correct “gen” to “gene”.

Author Response

Please see the attachment with our point-by-point response to your comments. Thanks. 

Reviewer 2 Report

This Review provides an overview of cancer-relevant DNA and RNA G-quadruplexes (G4s), as well as of DNA i-Motifs (iMs), focusing on the formation and function of these structures. Comments follow:

  1. In the Introduction, the authors state that “the topology of RNA G4s is limited to the parallel conformation” (p. 2, l. 56); while in the paragraph 2.2 they state that “a newly described antiparallel topology in TERRA has shown that G4s could also be polymorphic and adopt different structures apart from the canonical parallel configuration” (p. 9, l. 284). Please clarify this point.
  2. If it is true that RNA can adopt different structures apart from the canonical parallel configuration, Fig. 1B, which shows only a parallel RNA configuration, should be modified. Furthermore, the two G-tetrads in Fig. 1A and Fig. 1B are identical. Since it is redundant to show the same G-tetrad twice, please remove one.
  3. In the introduction, the authors also state: “it is widely held that G-rich RNA sequences are more prone to the formation of quadruplex structures” (p. 2, l. 51). However, Guo and Bartel showed that RNA G-quadruplexes are globally unfolded in eukaryotic cells (Science 2016, 353(6306), aaf5371). Please discuss about this point.
  4. In this reviewer’s opinion, the part dedicated to human telomeric G4s is too small compared to that dedicated to the other G4 structures.
  5. Biroccio and collaborators have provided evidence for the existence of (1) a G4 in the promoter of vegfr-2, the main VEGF-A receptor expressed on the surface of endothelial cells regulating tumor angiogenesis (Nucleic Acids Res. 2014, 42, 2945-57); and of (2) G4 structures within the CD133 gene (Nucleic Acids Res. 2016, 44, 1579-90). Being both cancer-relevant DNA G4s, this reviewer believes that it should be appropriate to add these findings to the manuscript.
  6. The authors use the terms "G-quartet(s)" and "G-tetrad(s)" indiscriminately. To avoid confusion in readers without experience in the study of G-quadruplex structures, this reviewer suggests using only one of the two terms. Please, don't forget the "G" before tetrad (p. 7, l. 198) or quartet (p. 13, l. 431).

Author Response

(The authors gave the same response as above.)

Round 2

Reviewer 2 Report

The manuscript has certainly been improved over to the previous version. Therefore, this reviewer believes that it can now be accepted.